# Long-Term Outcomes of a Randomized Study of Neoadjuvant Induction Dual HER2 Blockade with Trastuzumab and Lapatinib Followed by Weekly Paclitaxel Plus Dual HER2 Blockade for HER2-Positive Primary Breast Cancer (Neo-Lath Study)

**DOI:** 10.3390/cancers13164008

**Published:** 2021-08-09

**Authors:** Eriko Tokunaga, Norikazu Masuda, Naohito Yamamoto, Hiroji Iwata, Hiroko Bando, Tomoyuki Aruga, Shoichiro Ohtani, Tomomi Fujisawa, Toshimi Takano, Kenichi Inoue, Nobuyasu Suganuma, Masahiro Takada, Kenjiro Aogi, Kenichi Sakurai, Hideo Shigematsu, Katsumasa Kuroi, Hironori Haga, Shinji Ohno, Satoshi Morita, Masakazu Toi

**Affiliations:** 1Department of Breast Oncology, National Hospital Organization Kyushu Cancer Center, 3-1-1 Notame Fukuoka Minami-ku, Fukuoka-shi 811-1395, Fukuoka, Japan; tokunaga.eriko.pw@mail.hosp.go.jp; 2Department of Surgery, Breast Oncology, National Hospital Organization Osaka National Hospital, 2-1-14 Hoenzaka, Chuo-ku, Osaka-shi 540-0006, Osaka, Japan; 3Division of Breast Surgery, Chiba Cancer Center, 666-2 Nitona-cho, Chuo-ku, Chiba-shi 260-8717, Chiba, Japan; nyamamot@chiba-cc.jp; 4Department of Breast Oncology, Aichi Cancer Center Hospital, 1-1 Kanokoden, Chikusa-ku, Nagoya-shi 464-8681, Aichi, Japan; hiwata@aichi-cc.jp; 5Breast and Endocrine Surgery, Faculty of Medicine, University of Tsukuba, 1-1-1 Tennodai, Tsukuba-shi 305-8575, Ibaraki, Japan; bando@md.tsukuba.ac.jp; 6Department of Breast Surgery, Tokyo Metropolitan Cancer and Infectious Diseases Center Komagome Hospital, 18-22, Honkomagome 3-chome, Bunkyo-ku, Tokyo 113-8677, Japan; aruga@cick.jp; 7Division of Breast Surgery, Hiroshima City Hiroshima Citizens Hospital, 7-33 Motomachi, Naka-ku, Hiroshima-shi 730-8518, Hiroshima, Japan; info@ohtani-nyusen.jp; 8Department of Breast Oncology, Gunma Prefectural Cancer Center, 617-1 Takabayashi Nishimachi, Ohta-shi 373-8550, Gunma, Japan; fujisawa@gunma-cc.jp; 9Department of Medical Oncology, Toranomon Hospital, 2-2-2 Toranomon, Minato-ku, Tokyo 105-8470, Japan; takano@toranomon.gr.jp; 10Division of Breast Oncology, Saitama Cancer Center, 780 Komuro Inamachi, Kitaadachi-gun, Saitama 362-0806, Japan; ino@saitama-pho.jp; 11Department of Breast and Endocrine Surgery, Kanagawa Cancer Center, 2-3-2 Nakao, Asahi-ku, Yokohama-shi 241-8515, Kanagawa, Japan; suganuma@yokohama-cu.ac.jp; 12Breast Cancer Unit, Kyoto University Graduate School of Medicine, 54 Kawahara-cho, Shogoin Sakyo-ku, Kyoto-shi 606-8507, Kyoto, Japan; masahiro@kuhp.kyoto-u.ac.jp (M.T.); toi@kuhp.kyoto-u.ac.jp (M.T.); 13Department of Breast Oncology, National Hospital Organization Shikoku Cancer Center, 160 Kou Minamiumemotomachi, Matsuyama-shi 791-0280, Ehime, Japan; aogi.kenjiro.zx@mail.hosp.go.jp; 14Breast and Endocrine Surgery, Nihon University Itabashi Hospital, 30-1 Oyaguchikamicho Itabashi-ku, Tokyo 173-8610, Japan; sakurai.kenichi@tky.ndu.ac.jp; 15Department of Breast Surgery, National Hospital Organization Kure Medical Center and Chugoku Cancer Center, 3-1 Aoyamacho, Kure-shi 737-0023, Hiroshima, Japan; shigematsu.hideo.tf@mail.hosp.go.jp; 16Department of Breast Surgery, Tokyo Metropolitan Health and Hospitals Corporation Ebara Hospital, 4-5-10 Higashiyukigaya, Ota-ku, Tokyo 145-0065, Japan; kurochan@tokyo-hmt.jp; 17Department of Diagnostic Pathology, Kyoto University Hospital, 54 Kawahara-cho, Shogoin, Sakyo-ku, Kyoto-shi 606-8507, Kyoto, Japan; haga@kuhp.kyoto-u.ac.jp; 18Breast Oncology Center, The Cancer Institute Hospital of JFCR, 3-8-31 Ariake, Koto-ku, Tokyo 135-8550, Japan; shinji.ohno@jfcr.or.jp; 19Department of Biomedical Statistics and Bioinformatics, Kyoto University Graduate School of Medicine, 54 Kawahara-cho, Shogoin, Sakyo-ku, Kyoto-shi 606-8507, Kyoto, Japan; smorita@kuhp.kyoto-u.ac.jp

**Keywords:** anti-HER2 therapy, HER2-positive breast cancer, lapatinib, long-term prognosis, neoadjuvant chemotherapy, paclitaxel

## Abstract

**Simple Summary:**

We conducted the Neo-LaTH study in which patients with HER2-positive breast cancer were randomized to different lengths of neoadjuvant induction anti-HER2 therapy with lapatinib and trastuzumab followed by weekly paclitaxel plus anti-HER2 therapy, and in estrogen receptor-positive patients, with or without concurrent endocrine therapy. Here, we report the survival outcomes. The duration of neoadjuvant induction therapy and/or the addition of endocrine therapy at randomization did not affect the pathological complete response (CpCR) rate after neoadjuvant treatment and long-term outcomes. The 5-year disease-free survival rate was significantly higher in patients who had CpCR plus ypN0 after neoadjuvant treatment than in those who did not (91.7% vs. 85.1%; *p* = 0.0387). The stratified analysis showed better survival outcomes in CpCRypN0 patients than non-CpCRypN0 patients, regardless of use of adjuvant anthracycline therapy. Favorable survival outcomes, regardless of adjuvant anthracycline, were particularly noted in patients with small size and clinically node-negative tumors.

**Abstract:**

We conducted the Neo-LaTH study in which patients were randomized to different lengths of neoadjuvant induction anti-HER2 therapy with lapatinib and trastuzumab followed by weekly paclitaxel plus the anti-HER2 therapy, and in estrogen receptor (ER)-positive patients, with or without concurrent endocrine therapy. The use of endocrine therapy did not affect the response; comprehensive pathological complete response (CpCR) plus ypN0 rate was 57.6% and 30.3% in ER-negative and ER-positive patients, respectively. After surgery, patients received an anthracycline-based regimen based on physician’s choice, followed by trastuzumab for 1 year, and in ER-positive patients, endocrine therapy for 5 years. Here, we report the 5-year survival outcomes. Among the followed-up patients (*n* = 212), the 5-year disease-free survival (DFS), distant DFS, and overall survival rates were 87.8% [95% confidence interval (CI), 82.5–91.6%], 93.7% (95% CI, 89.3–96.3%), and 95.6% (95% CI, 91.7–97.7%), respectively, with no difference between ER-negative and ER-positive patients. The 5-year DFS rate was significantly higher in patients who had a CpCR plus ypN0 after neoadjuvant treatment than in those who did not (91.7% vs. 85.1%; *p* = 0.0387). The stratified analysis showed better survival outcomes in patients who had CpCRypN0 than in those who did not after neoadjuvant treatment, regardless of use of adjuvant anthracycline therapy.

## 1. Introduction

Human epidermal growth factor receptor 2 (HER2)-positive breast cancer is an aggressive phenotype associated with a poor prognosis [1]. However, with development of HER2-targeted therapy, such as the humanized monoclonal antibody trastuzumab, the prognosis of HER2-positive breast cancer has markedly improved [2,3,4,5,6]. Combination therapy of two HER2 targeted drugs can be applied to avoid drug resistance to a single agent while expecting synergistic effects. Trastuzumab-containing dual HER2 blockade therapy has been shown to produce a greater survival benefit compared with trastuzumab alone [7]. In the neoadjuvant setting for early-stage, HER2-positive breast cancer, several studies have reported increased efficacy by adding dual HER2 blockade therapy to chemotherapy [8,9,10]. The NeoSphere study [11] investigated blockade with two monoclonal antibodies, trastuzumab and pertuzumab. In the NeoALTTO study [12], a combination of lapatinib, which is a small-molecule tyrosine kinase inhibitor, and trastuzumab was examined. Both of these studies reported a significantly increased pathological complete response (pCR) rate.

We previously reported the results of the randomized phase II Neo-LaTH study (JBCRG-16) [13]. In this study, patients with HER2-positive primary breast cancer (T1c-3 N0-1 M0; target lesion ≤ 7 cm) were randomized to different lengths of neoadjuvant induction anti-HER2 therapy with lapatinib and trastuzumab followed by weekly paclitaxel plus anti-HER2 therapy, and in estrogen receptor (ER)-positive patients, with or without endocrine therapy; the primary endpoint was comprehensive pCR (CpCR) rate, including residual ductal carcinoma in situ of the breast (ypT0 or Tis). Of the 212 patients enrolled, 101 (47.9%) had a CpCR. The CpCR rate was significantly higher in ER-negative patients than in ER-positive patients (63.0% vs. 36.1%; *p* = 0.0034).

Here, we report the survival outcomes of patients who were enrolled in the Neo-LaTH study. We also report the findings from subgroup analyses stratified by the response to neoadjuvant treatment and use of adjuvant chemotherapy.

## 2. Materials and Methods

### 2.1. Trial Design

The Neo-LaTH study was conducted between March 2012 and September 2013 in 16 centers in Japan. For the present follow-up study, data were analyzed from 8 July 2019 to 21 October 2020. Details of the Neo-LaTH study have been published previously [13]. In brief, this was a randomized, phase II, five-arm study that evaluated the efficacy and safety of neoadjuvant induction anti-HER2 therapy with lapatinib and trastuzumab followed by anti-HER2 therapy plus weekly paclitaxel with or without prolongation of anti-HER2 therapy before surgery in patients with HER2-positive and ER-positive or ER-negative breast cancer. Appendix A shows the study design. Patients were classified into 5 groups according to their ER status: ER-negative patients were randomized to groups A and B, and ER-positive patients to groups C, D, and E. Patients in groups A, C, and D received lapatinib and trastuzumab for 6 weeks, and those in groups B and E received lapatinib and trastuzumab for 18 weeks, followed by lapatinib and trastuzumab plus weekly paclitaxel for 12 weeks. Patients in groups D and E also received endocrine therapy. The study was registered at http://www.umin.ac.jp/ctr/index-j.htm (UMIN000007576; released on 26 March 2012; accessed on 16 June 2021; last updated on 16 June 2021).

After surgery, patients received an anthracycline-based regimen depending on the physician’s choice and response to neoadjuvant treatment (this regimen could be omitted in cases of pCR and N0). The anthracycline-based therapy comprised four cycles of an FEC100 (5-fluorouracil 500 mg/m^2^, epirubicin 100 mg/m^2^, cyclophosphamide 500 mg/m^2^) or AC (doxorubicin and cyclophosphamide) regimen. Subsequently, patients received trastuzumab (initial dose of 8 mg/kg followed by 6 mg/kg, every 3 weeks) for at least 52 weeks. In ER-positive patients, standard endocrine therapy was administered for 5 years after surgery, regardless of the length of neoadjuvant endocrine therapy. Radiation therapy was also administered postoperatively after completion of the anthracycline therapy.

### 2.2. Patients

All patients who participated in the Neo-LaTH study were included in this follow-up study, except for those who withdrew consent for follow-up or those who died. Details of inclusion/exclusion criteria were described previously [13].

### 2.3. Observation and Endpoints

The following items were evaluated at 6 months and 1, 2, 3, 4, and 5 years after surgery (±1 month): metastasis or recurrence, development of secondary cancer, survival, details of treatment provided after surgery, and outcomes of adverse events requiring follow-up.

Endpoints included DFS, DDFS, and OS. DFS was defined as the period from the date of study enrollment to death of any cause, recurrence of primary breast cancer, or event of secondary cancer. DDFS was defined as the period from the date of study enrollment to diagnosis of distant metastasis of the primary cancer. OS was defined as the period from the registration date to death of any cause.

The results were stratified by treatment response (with or without CpCRypN0) to neoadjuvant treatment (i.e., trastuzumab, lapatinib, and paclitaxel), as previously reported [13]. CpCRypN0 was defined as the absence of residual invasive tumor in the breast and lymph node metastasis in sentinel node biopsy and/or dissection performed after systemic treatment (even if the absence of sentinel lymph node metastasis was confirmed before starting therapy). The results were also stratified by use of adjuvant anthracycline therapy for exploratory purposes.

New breast cancer was not considered as an event in this analysis. New breast cancers were classified into other types (i.e., ipsilateral or contralateral, invasive or noninvasive cancer, and ductal carcinoma in situ).

### 2.4. Statistical Analysis

For DFS, DDFS, and OS, the Kaplan–Meier method was used to estimate survival curves, and the log-rank test was used for comparisons between groups. To analyze treatment effects, HRs and 95% Cis were calculated for each group using the Cox proportional hazards model. Statistical analyses were carried out using SAS version 9.4 (SAS Institute Inc., Cary, NC, USA).

## 3. Results

### 3.1. Patients

The Neo-LaTH study was conducted between March 2012 and September 2013 in 16 centers in Japan. For the follow-up study, the data cut-off was 8 July 2019, and the data were fixed on 21 October 2020.

All patients (*n* = 212) who participated in the Neo-LaTH study were included in the present follow-up study; although 1 patient (group E) withdrew during the follow-up period, this patient provided consent to use her data on survival from randomization to the date of consent withdrawal, and these data were included in the analysis (Appendix A).

Table 1 shows the patients’ characteristics. The median age was 53 years (range: 26–70 years) and the duration of follow-up was 2074 days (range: 63–2425 days). CpCRypN0 rate was 65.9% and 58.3% in groups A and B (ER-negative cohort), and 31.7%, 33.3%, and 37.5% in groups C–E (ER-positive cohort), showing no difference within each cohort associated with the duration of neoadjuvant induction therapy and/or the addition of endocrine therapy. Of the 130 patients who underwent breast conservation surgery, 5 patients (3.8%) did not receive postoperative radiation therapy; although the reason was not specified in the case report form, it may be because of the patient’s request or refusal. 

### 3.2. Survival Outcomes

The 5-year disease-free survival (DFS) rate was 87.8% (95% confidence interval [CI], 82.5–91.6). Although DFS rate was slightly higher in the ER-positive cohort than in the ER-negative cohort at 3 years, no difference was found at 5 years (Figure 1). The 5-year distant disease-free survival (DDFS) rate was 93.7% (95% CI, 89.3–96.3) (Figure 2). The 5-year overall survival (OS) rate was 95.6% (95% CI, 91.7–97.7) (Figure 3). There seems to be no significant difference between ER-negative and ER-positive cohorts in terms of DDFS and OS.

By subgroups at randomization, the 5-year DFS was 88.3% and 89.4% in groups A and B (ER-negative cohort), and 85.4%, 86.9% and 89.3% in groups C–E (ER-positive cohort). The 5-year DDFS was 95.3% and 93.7% in groups A and B (ER-negative cohort), and 90.2%, 94.7%, and 94.6% in groups C–E (ER-positive cohort). The 5-year OS was 97.6 and 93.5%, and 97.6%, 92.1%, and 97.3% in groups C–E (ER-positive cohort). Therefore, no substantial difference in the survival outcomes was noted in both cohorts associated with the duration of neoadjuvant induction therapy and/or the addition of endocrine therapy.

### 3.3. Subgroup Analyses

#### 3.3.1. Stratified by Response to Neoadjuvant Treatment

The 5-year DFS rate was significantly higher in patients who had CpCRypN0 (i.e., CpCR with a pathologically negative axilla) after neoadjuvant treatment than those who did not (91.7% vs. 85.1%; *p* = 0.0387) (Appendix A). In the ER-negative cohort, the 5-year DFS rate tended to be better in patients who had CpCRypN0, although no significant difference was found (Appendix A). In the ER-positive cohort, there was no difference in the 5-year DFS rate between patients with and without CpCRypN0 (Appendix A). 

The 5-year DDFS rate was 97.9% and 90.7% in patients with and those without CpCRypN0, respectively. There tended to be a better outcome of 5-year DDFS in patients with CpCRypN0 in the ER-negative cohort, whereas in the ER-positive cohort, no difference was found between patients with and without CpCRypN0.

The 5-year OS rate was 97.9% and 93.5% in patients with and those without CpCRypN0, respectively. The difference in the 5-year OS rate between patients with and those without CpCRypN0 was 7.1% in the ER-negative cohort and 2.8% in the ER-positive cohort.

#### 3.3.2. Stratified by Response to Neoadjuvant Treatment and by Use of Adjuvant Anthracycline Therapy

The results were then stratified by response to neoadjuvant treatment (with or without CpCRypN0) and use of adjuvant anthracycline therapy (physician’s choice). A total of 48.6% (102/210) of patients received adjuvant anthracycline therapy. A total of 35.9% (33/92) of patients in the ER-negative cohort and 58.5% (69/118) of patients in the ER-positive cohort received adjuvant anthracycline therapy (Table 2). Overall, in patients with CpCRypN0, a good DDFS was observed in patients with and those without use of adjuvant anthracycline therapy. In patients without CpCRypN0, the DDFS rate tended to be lower than that observed in patients with CpCRypN0 in both patients with and without use of adjuvant anthracycline therapy (Figure 4). A similar tendency was observed in the ER-negative cohort. However, in the ER-positive cohort, no difference in DDFS rate was found between patients with and without use of adjuvant anthracycline therapy in both patients with and without CpCRypN0.

### 3.4. Exploratory Analysis

Patients with cancer in the early stage (T1cT2N0) were also analyzed; patients were stratified by response to neoadjuvant treatment (with or without CpCRypN0) and use of adjuvant anthracycline therapy in patients with cancer stage T1cT2N0. The 5-year DDFS rate was 100% in all groups of patients, except for those who did not have CpCRypN0, but received adjuvant anthracycline (94.3%) (Appendix A). A similar trend was observed for ER-negative and ER-positive cohorts.

## 4. Discussion

### 4.1. Survival Outcomes

In this long-term follow-up study of patients receiving neoadjuvant dual anti-HER2 therapy with lapatinib and trastuzumab combined with weekly paclitaxel, we found the following main findings: The 5-year DFS, DDFS, and OS rates were approximately 90%. In subgroup analysis of stratified patients by response to neoadjuvant treatment, survival outcomes were better in patients who achieved CpCRypN0 than in those who did not. A tendency for a good prognosis in patients who achieved CpCRypN0 was also found in the ER-negative cohort.

The 5-year DFS rate was 87.8% (95% CI, 82.5–91.6) in the present study. Previously, the NeoSphere study investigated pertuzumab and trastuzumab plus docetaxel in various combinations in a neoadjuvant setting [11]. Follow-up in this study showed that dual anti-HER2 therapy with pertuzumab and trastuzumab plus docetaxel was associated with improved long-term outcomes (5-year progression-free survival rate, 86% [95% CI, 77–91]; 5-year DFS rate, 84% [95% CI, 72–91]). Additionally, patients who achieved a total pCR had a longer progression-free survival rate than those who did not (85% vs. 76%; hazard ratio [HR], 0.54) [14]. The NeoALTTO study, which is of similar design to the present study, investigated lapatinib and trastuzumab combined with weekly paclitaxel [12]; the 3-year event-free survival rate was significantly improved in patients with a pCR than in those who did not (86% vs. 72%; HR, 0.38), as was the 3-year OS rate (94% vs. 87%; HR, 0.35) [15]. An updated 6-year analysis also showed a significantly better event-free survival rate (77% vs. 65%; HR, 0.54) and OS rate (89% vs. 77%; HR, 0.43) in patients who had a pCR than in those who did not [16]. Notably, the benefit observed in patients with a pCR was higher in hormone receptor-negative patients than in hormone receptor-positive patients. The association of a pCR with long-term clinical benefit has also been shown in pooled data from clinical trials of neoadjuvant treatment [17,18]. Our finding that patients who had CpCRypN0 had better survival outcomes than those who did not is consistent with these findings.

The role of hormone receptor status in the prognosis of HER2-positive breast cancer remains controversial. Several studies that investigated dual HER2-blockade as adjuvant therapy for patients with HER2-positive early breast cancer reported contradictory findings. In the ExteNET trial, patients who completed neoadjuvant and adjuvant chemotherapy plus trastuzumab received either neratinib or placebo for 1 year; an improvement in the 5-year invasive DFS with neratinib was observed in the hormone receptor-positive group, but only a transient effect that diminished after stopping treatment was observed in the hormone receptor-negative group [19]. The ALTTO study investigated chemotherapy plus adjuvant trastuzumab and/or lapatinib and showed that hormone receptor-positive patients had better survival outcomes in the first 5 years than hormone receptor-negative patients. However, survival outcomes at 8 years became similar [20,21]. The APHINITY study reported an improved outcome in hormone receptor-negative patients who received adjuvant trastuzumab plus pertuzumab [22].

The observed outcomes may be interpreted in relation to the degree of dependence of the tumor in the HER2 signaling pathway. HER2-positive/hormone receptor-negative tumors tend to be more aggressive and highly dependent on HER2 signaling. In this population, there is a high risk of recurrence relatively shortly after surgery (generally, within 2–3 years). The prognostic value of pCR is greatest in aggressive cancer subtypes, including HER2-positive, hormone receptor-negative breast cancer [18]. Therefore, achieving a pCR after neoadjuvant therapy may be particularly important for this subtype. Our results also showed an improved outcome in patients with pCR in the early period (2–3 years after surgery), but they remained unchanged thereafter in the ER-negative cohort. By contrast, HER2-positive/hormone receptor-positive tumors are generally less dependent on HER2 signaling than HER2-positive/hormone receptor-negative tumors. HER2-positive/hormone receptor-positive tumors cause upregulation of hormone receptor signaling as an adaptive mechanism of cell survival, which may lead to partial resistance to anti-HER2 therapy [23]. In this subtype, risk of recurrence remains for a longer time. The crosstalk between ER and HER2 receptor signaling may be an important contributor to the development of resistance to therapies against the ER pathway [24]. It is suggested that concurrent targeting of ER and HER2 may improve treatment efficacy and overcome ER-mediated resistance. Clinically, the NRG Oncology/NSABP B-52 study showed that the addition of endocrine therapy to neoadjuvant therapy (docetaxel, carboplatin, trastuzumab, and pertuzumab) improved pCR rates numerically, but the improvement was not statistically significant [25]. Likewise, in the present study, CpCR rate after neoadjuvant treatment and long-term survival outcomes were similar among ER-positive patients, regardless of the addition of endocrine therapy to neoadjuvant anti-HER2 therapy plus paclitaxel at randomization. Moreover, generally, a pCR can be an important prognostic factor for luminal type B breast cancers, whereas the effect of neoadjuvant chemotherapy is not a good indicator for luminal type A [26,27,28]. A mixture of these two types may explain why there is little difference in prognosis between patients with and without a pCR in this cohort.

### 4.2. Omission of Adjuvant Anthracycline Therapy

In the present study, 108 (51.4%) patients did not receive adjuvant anthracycline therapy. The decision was made by attending physicians mainly depending on the response to neoadjuvant treatment, and in some patients, based on the patient’s preference. The ad-hoc stratified analysis showed better survival outcomes in patients who had CpCRypN0 than in those who did not, regardless of use of adjuvant anthracycline therapy. The tendency of better outcomes in patients with CpCRypN0 in both with and without use of adjuvant anthracycline therapy was also noted in the ER-negative cohort. We believe that these findings are novel, as in the previous NeoSphere and Neo-ALTTO studies, all patients received anthracycline (three cycles of FEC: 5-fluorouracil–epirubicin–cyclophosphamide) regimen after surgery [11,12]; therefore, the possibility of omission of adjuvant anthracycline therapy was not examined. However, because the decision on use of adjuvant anthracycline therapy was made in a real-world setting, and a potential selection bias cannot be ruled out, the results should be interpreted with caution.

It was also found that, among patients who did not achieve CpCRypN0 after neoadjuvant treatment, a smaller proportion of patients in ER-negative cohort received adjuvant anthracycline: 45.7% (16/35) and 71.4% (55/77) in ER-negative and ER-positive cohorts, respectively. The result appears unexpected, but it may be because the decision of the attending physician was made taking account of the volume of residual invasive tumor (tumor diameter and residual cancer burden) as well as presence or absence of axillary lymph node metastasis. In fact, among patients who did not achieve CpCRypN0, the percentage of patients with residual invasive tumor diameter <1 cm plus pN0 was 16 of 35 (45.7%) in the ER-negative cohort whereas it was 22 of 77 (28.6%) in the ER-positive cohort, indicating that many of the ER-negative HER2-positive patients who did not achieve CpCRypN0 had minimal residual disease.

Regarding the possibility of an anthracycline-free neoadjuvant chemotherapy regimen, we previously investigated this in the setting of single HER2 blockade with trastuzumab for the treatment of HER2-positive breast cancer (JBCRG-10 study) [29]. This study was originally designed to investigate different sequences of chemotherapy (anthracycline-first or taxane-first regimen). However, after one death in the anthracycline-containing arm, all patients received the docetaxel–cyclophosphamide–trastuzumab regimen without anthracycline. This regimen resulted in a good prognosis, particularly in patients with early-stage disease. An anthracycline-free regimen was also examined in the BCIRG-006 study, which compared 3 regimens: doxorubicin and cyclophosphamide followed by docetaxel (AC-T) regimen, AC-T plus trastuzumab, and docetaxel and carboplatin plus trastuzumab. While the efficacy was similar between the two trastuzumab-containing regimens, the non-anthracycline regimen had fewer acute toxic effects, and lower risks of cardiotoxicity and leukemia [30].

Because of concerns for anthracycline-induced cardiac toxicity, many physicians are willing to avoid anthracycline-containing regimens in patients who achieve a pCR or, even in patients who do not have a pCR, when the residual tumor is small (especially pN0 cases), as observed for patients in the ER-negative cohort in the present study. Our findings suggested a possible omission of anthracycline in patients who achieve CpCRypN0 after neoadjuvant treatment. However, a prospective, randomized study is necessary to validate these results. Moreover, in patients who had residual invasive tumor after neoadjuvant treatment, the survival outcomes were less favorable in both cases with and without adjuvant anthracycline therapy. In such cases, more intensive adjuvant chemotherapy regimens may be needed; novel antibody-drug conjugate agents including trastuzumab emtansine (T-DM1) [31] and trastuzumab deruxtecan (DS-8201a) [32,33] are being investigated and promising results have been obtained.

### 4.3. Outcomes in Patients with Small Node-Negative Tumor

In the present study, 106 patients had small node-negative tumor (T1cT2N0) and showed favorable survival outcomes, regardless of use of adjuvant anthracycline therapy. The 5-year DDFS rate in patients with T1cT2N0 disease ranged from 94.3% to 100%. Moreover, ad-hoc analysis showed that no event occurred in patients with CpCRypN0, whereas among non-CpCRypN0 patients, one patient with an ER-negative tumor had lung metastasis and one patient with an ER-positive tumor had brain metastasis.

Previously, the APT trial investigated the adjuvant paclitaxel-trastuzumab regimen in HER2-positive patients with a stage I/II tumor (tumor < 3 cm; N0), and showed positive results [34,35]. These findings suggest that omission of adjuvant anthracycline therapy may be an option in patients with small, node-negative tumors, especially in patients who had CpCRypN0 after neoadjuvant therapy. Because patients with small tumors are generally not included in randomized studies, further studies are required to determine individualized, less toxic, adjuvant therapy.

### 4.4. Study Limitations

This study is limited by a small sample size. Also, the decision on use of adjuvant anthracycline therapy was made by attending physicians in a real-world setting, and not in a randomized manner, which might have caused selection bias. Therefore, the results should be interpreted with caution. Furthermore, the dual anti-HER2 therapy with lapatinib and trastuzumab is not a current standard neoadjuvant therapy for primary HER2-positive breast cancer, so caution is needed in applying the findings from the present study to clinical practice.

## 5. Conclusions

Neoadjuvant induction dual HER2 blockade therapy with trastuzumab and lapatinib plus paclitaxel resulted in significantly higher 5-year DFS rate in patients who achieved CpCRypN0 than in those who did not after neoadjuvant treatment. Omission of adjuvant anthracycline therapy may be considered in patients who achieved CpCRypN0 after neoadjuvant treatment. Further large-scale studies with a longer follow-up are required to confirm the clinical utility of this regimen in patients with HER2-positive ER-positive or ER-negative primary breast cancer.

## Figures and Tables

**Figure 1 cancers-13-04008-f001:**
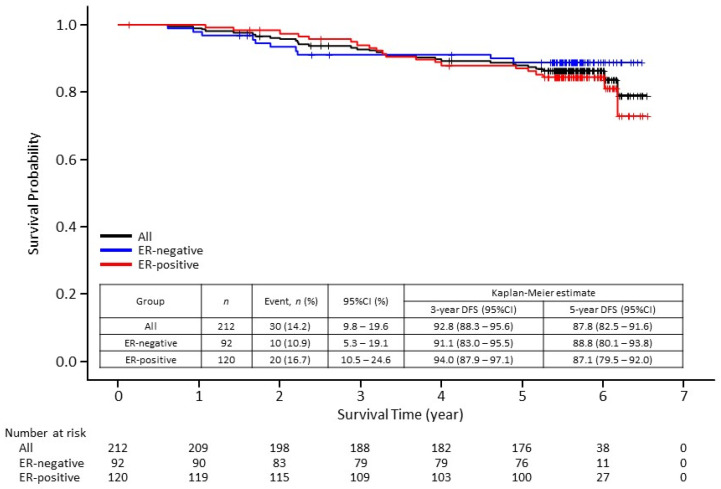
Kaplan–Meier curves of disease-free survival (DFS) in 212 patients enrolled in the Neo-LaTH study comprising 92 estrogen receptor (ER)-negative and 120 ER-positive patients. CI, confidence interval.

**Figure 2 cancers-13-04008-f002:**
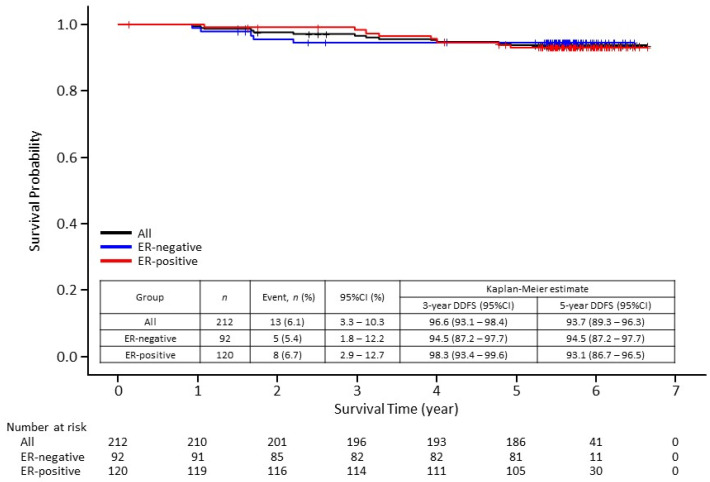
Kaplan–Meier curves of distant disease-free survival (DDFS) in 212 patients enrolled in the Neo-LaTH study comprising 92 estrogen receptor (ER)-negative and 120 ER-positive patients. CI, confidence interval.

**Figure 3 cancers-13-04008-f003:**
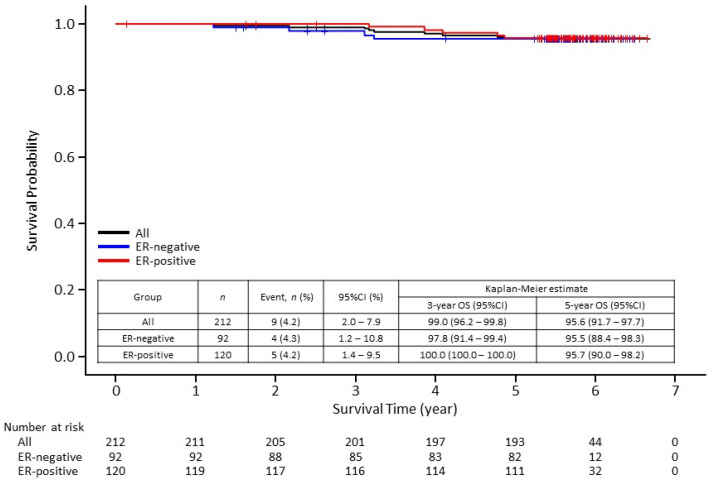
Kaplan–Meier curves of overall survival (OS) in 212 patients enrolled in the Neo-LaTH study comprising 92 estrogen receptor (ER)-negative and 120 ER-positive patients. CI, confidence interval.

**Figure 4 cancers-13-04008-f004:**
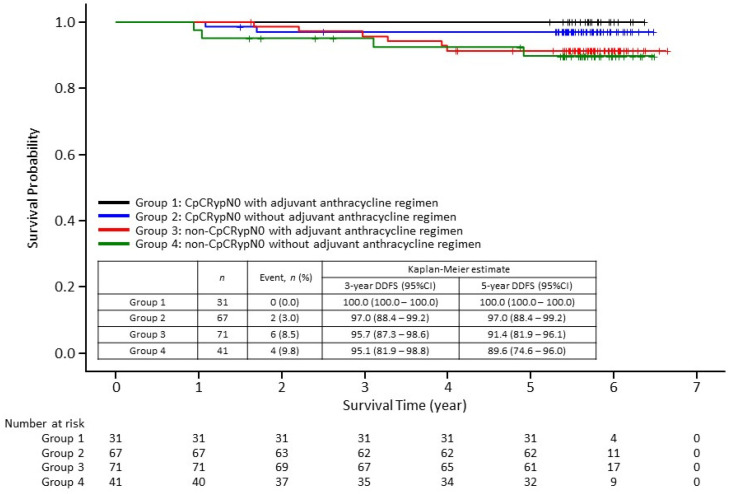
Kaplan–Meier curves of distant disease-free survival (DDFS) stratified by response to neoadjuvant treatment (with or without CpCRypN0) and with or without use of adjuvant anthracycline in all patients (*n* = 210). CI, confidence interval; CpCRypN0, comprehensive pathological complete response with a pathologically negative axilla.

**Table 1 cancers-13-04008-t001:** Patient characteristics (*n* = 212).

Characteristics		Group A *n* = 44	Group B *n* = 48	Group C *n* = 41	Group D *n* = 39	Group E *n* = 40	All *n* = 212
Age, years	Median	56	56	52	53	49	53
	Range	33–69	36–69	32–70	26–66	28–68	26–70
TNM staging before treatment start							
T: primary lesion	cT1	4 (9.1)	6 (12.5)	11 (26.8)	8 (20.5)	11 (27.5)	40 (18.9)
	T2	31 (70.5)	29 (60.4)	26 (63.4)	26 (66.7)	26 (65.0)	138 (65.1)
	T3	9 (20.5)	13 (27.1)	4 (9.8)	5 (12.8)	3 (7.5)	34 (16.0)
N: regional lymph node	N0	22 (50.0)	26 (54.2)	23 (56.1)	22 (56.4)	24 (60.0)	117 (55.2)
	N1	22 (50.0)	22 (45.8)	18 (43.9)	17 (43.6)	16 (40.0)	95 (44.8)
Histological grade (B&R)	1	0	2 (4.2)	1 (2.4)	0	0	3 (1.4)
	2	6 (13.6)	7 (14.6)	13 (31.7)	10 (25.6)	13 (32.5)	49 (23.1)
	3	11 (25.0)	11 (22.9)	9 (22.0)	12 (30.8)	8 (20.0)	51 (24.1)
	Unknown	27 (61.4)	28 (58.3)	18 (43.9)	17 (43.6)	19 (47.5)	109 (51.4)
Lymph node metastasis after surgery	pN0	40 (90.9)	44 (91.7)	33 (80.5)	31 (79.5)	32 (80.0)	180 (84.9)
	pN (+)	4 (9.1)	2 (4.2)	7 (17.1)	6 (15.4)	5 (12.5)	24 (11.3)
	Unknown	0	2 (4.2)	1 (2.4)	2 (5.1)	3 (7.5)	8 (3.8)
Response to neoadjuvant chemotherapy (CpCRypN0)	Yes	29 (65.9)	28 (58.3)	13 (31.7)	13 (33.3)	15 (37.5)	98 (46.2)
	No	15 (34.1)	20 (41.7)	28 (68.3)	26 (66.7)	24 (60.0)	113 (53.3)
	Unknown	0	0	0	0	1 (2.5)	1 (0.5)
Surgical procedure	Breast-conserving surgery	28 (63.6)	26 (54.2)	29 (70.7)	21 (53.8)	26 (65.0)	130 (61.3)

	Total mastectomy	16 (36.4)	21 (43.8)	12 (29.3)	18 (46.2)	12 (30.0)	79 (37.3)
	Did not undergo surgery	0	1 (2.1)	0	0	2 (5.0)	3 (1.4)
Axillary dissection procedure	Axillary dissection	21 (47.7)	19 (39.6)	16 (39.0)	12 (30.8)	13 (32.5)	81 (38.2)

	Axillary sampling dissection	2 (4.5)	5 (10.4)	4 (9.8)	1 (2.6)	3 (7.5)	15 (7.1)
	SLN biopsy	21 (47.7)	23 (47.9)	21 (51.2)	26 (66.7)	22 (55.0)	113 (53.3)
	Did not undergo surgery	0	1 (2.1)	0	0	2 (5.0)	3 (1.4)

Postoperative radiation ^a^							
Patients undergoing breast conservation (*N* = 130)	Yes (with regional lymph node irradiation)	3 (6.8)	1 (2.1)	2 (4.9)	0	1 (2.5)	7 (3.3)
	Yes (without regional lymph node irradiation)	23 (52.3)	24 (50.0)	26 (63.4)	21 (53.8)	24 (60.0)	118 (55.7)
	No	2 (4.5)	1 (2.1)	1 (2.4)	0	1 (2.5)	5 (2.4)
Surgical procedure: Total mastectomy (*N* = 79)	Yes (with regional lymph node irradiation)	4 (9.1)	6 (12.5)	2 (4.9)	3 (7.7)	0	15 (7.1)
	Yes (without regional lymph node irradiation)	0	0	1 (2.4)	1 (2.6)	0	2 (0.9)
	No	12 (27.3)	15 (31.3)	9 (22.0)	14 (35.9)	12 (30.0)	62 (29.2)
Adjuvant chemotherapyAnthracycline ^b^	Yes	18 (40.9)	15 (31.3)	24 (58.5)	23 (59.0)	22 (55.0)	102 (48.1)
	No	26 (59.1)	33 (68.8)	17 (41.5)	16 (41.0)	17 (42.5)	109 (51.4)
Endocrine therapy ^b^	Yes	11 (25.0)	6 (12.5)	38 (92.7)	37 (94.9)	35 (87.5)	127 (59.9)
	No	33 (75.0)	42 (87.5)	3 (7.3)	2 (5.1)	4 (10.0)	84 (39.6)
ER status at registration	Negative	44 (100.0)	48 (100.0)	0	0	0	92 (43.4)
(central assessment)	Positive	0	0	41 (100.0)	39 (100.0)	40 (100.0)	120 (56.6)
	Positive (1–9%)	0	0	4 (9.8)	5 (12.8)	10 (25.0)	19 (9.0)
	Positive (≥10%)	0	0	37 (90.2)	34 (87.2)	30 (75.0)	101 (47.6)
Hormonal status of postoperative residual tumor cells in the breast ^c^	Not performed	0	6 (40.0)	3 (13.0)	2 (8.3)	2 (11.8)	13 (14.8)
	HR−	7 (77.8)	3 (20.0)	2 (8.7)	0	1 (5.9)	13 (14.8)
	HR+	2 (22.2)	6 (40.0)	18 (78.3)	22 (91.7)	14 (82.4)	62 (70.5)
HER2 status before treatment start	IHC3+	43 (97.7)	45 (93.8)	37 (90.2)	33 (84.6)	37 (92.5)	195 (92.0)
	IHC2 + DISH+	1 (2.3)	3 (6.3)	4 (9.8)	6 (15.4)	3 (7.5)	17 (8.0)
HER2 status of postoperative residual tumor cells in the breast ^c^	Not performed	0	6 (40.0)	6 (26.1)	5 (20.8)	3 (17.6)	20 (22.7)
	IHC3+	6 (66.7)	6 (40.0)	8 (34.8)	12 (50.0)	8 (47.1)	40 (45.5)
	IHC2 + FISH+	2 (22.2)	0	3 (13.0)	2 (8.3)	2 (11.8)	9 (10.2)
	IHC2 + FISH unknown	0	1 (6.7)	1 (4.3)	3 (12.5)	1 (5.9)	6 (6.8)
	IHC1+/0	1 (11.1)	2 (13.3)	5 (21.7)	2 (8.3)	3 (17.6)	13 (14.8)

Data are shown as n (%). ^a^ Three patients who did not undergo surgery (1 in group B and 2 in group E) were excluded. ^b^ One patient who withdrew consent during the follow-up period (group E) was excluded. ^c^ The percentage was calculated using the number of patients without QpCR (in the breast) as the denominator. B&R, Bloom and Richardson grading system; ER, estrogen receptor; HER2, human epidermal growth factor receptor 2; SLN, sentinel lymph node.

**Table 2 cancers-13-04008-t002:** Proportion of patients who received adjuvant anthracycline chemotherapy.

ER Status	Achieved CpCRypN0	Residual Invasive Disease	Total
ER-negative	29.8% (17/57)	45.7% (16/35)	35.9% (33/92)
ER-positive	34.1% (14/41)	71.4% (55/77)	58.5% (69/118)
Total	31.6% (31/98)	63.4% (71/112)	48.6% (102/210 ^a^)

ER, estrogen receptor. ^a^ From the total 212 patients, one patient who withdrew consent during the follow-up period (group E) and 1 patient who did not undergo surgery (group E) were excluded.

## Data Availability

Data available on request due to privacy/ethical restrictions.

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
