# Peer review of "Long-Term Outcomes of a Randomized Study of Neoadjuvant Induction Dual HER2 Blockade with Trastuzumab and Lapatinib Followed by Weekly Paclitaxel Plus Dual HER2 Blockade for HER2-Positive Primary Breast Cancer (Neo-Lath Study)"

_cancers, 2021, doi:10.3390/cancers13164008_

Round 1

Reviewer 1 Report

1) I found the title misleading - it implies that the study randomized patients to endocrine therapy or not. It should simply state that you are presenting the long-term outcomes for a randomized study of neoadjuvant induction trastuzumab and lapatinib followed by weekly paclitaxel with dual HER2 blockade. 

2) Similarly, the description of the study in the Abstract is confusing. You randomized patients to different lengths of induction dual HER2 blockade, then within the ER+ stratum, to concurrent endocrine therapy or not.

3) Results in table 1 suggest that the duration of induction therapy and the addition of endocrine did not impact pCR rates in the two strata but you should say that explicitly. Also, you should state explicitly that these randomizations did not impact DFS or other long-term outcomes (assuming this is the case). 

4) In Table 1 you should break down receipt of post-op radiation separately between the BCS vs. mastectomy patients; as presented it looks like only half on the BCS patients received radiation. Also, in that same table, what does QpCRypN0 refer to? Actually, aside from the pCR rates, the rest of the table could be moved to the Supplement, as the data is really not necessary for your current analysis. 

5) The absence of a significant difference in DFS between pCR and non-pCR among the ER- cohort is, presumably, a result of the small size of the cohort - would cite statistics and eliminate Figure 4. Similarly, once you've shown results for DFS, you can simply cite statistics for DDFS and OS - Figures 5 and 6 are unnecessary (again you could put these in the Supplement but they don't improve the manuscript).

6) Table 2 is confusing - would revise or eliminate and simply state the percentages of patients who did or did not receive adjuvant chemotherapy by cohort and pCR, especially since it didn't make a difference . Would keep Figure 7A but eliminate 7B and 7C. 

7) Would mention outcomes in the lower risk subgroup but eliminate Figure 8. 

8) in your discussion of outcomes in the ER+ cohort and the role of 'crosstalk' in determining response to neoadjuvant therapy +/- endocrine therapy, you should mention the negative results from NSABP B-52.

9) You correctly point out that the ability to draw conclusions from the lack of benefit from adjuvant chemotherapy is limited since this was not randomized, but your data do not support your conclusion that it can be omitted only in patients who achieve pCR.  

Author Response

Manuscript ID: cancers-1289943

Long-term Outcomes of a Randomized Study of Neoadjuvant Induction Dual HER2 Blockade with Trastuzumab and Lapa-tinib Followed by Weekly Paclitaxel plus Dual HER2 Blockade for HER2-Positive Primary Breast Cancer (Neo-Lath Study)

Dear Reviewer,

Thank you for considering the above manuscript for publication in Cancers. Your comments were highly insightful and enabled us to improve the quality of the manuscript. In the following pages, please find our point-by point response to each of the comments.

In the manuscript file, revisions are shown using the Microsoft Word tracked changes feature. We trust that the revisions in the manuscript will be sufficient to make our manuscript suitable for publication in Cancers.

-----------------------------------------------

Comments from Reviewer 1

1) I found the title misleading - it implies that the study randomized patients to endocrine therapy or not. It should simply state that you are presenting the long-term outcomes for a randomized study of neoadjuvant induction trastuzumab and lapatinib followed by weekly paclitaxel with dual HER2 blockade. 

Response

As suggested, we have revised the title as follows.

“Long-term Outcomes of a Randomized Study of Neoadjuvant Induction Dual HER2 Blockade with Trastuzumab and Lapatinib Followed by Weekly Paclitaxel plus Dual HER2 Blockade for HER2-Positive Primary Breast Cancer (Neo-Lath Study)”

2) Similarly, the description of the study in the Abstract is confusing. You randomized patients to different lengths of induction dual HER2 blockade, then within the ER+ stratum, to concurrent endocrine therapy or not.

Response

The abstract was revised as follows.

“We conducted the Neo-LaTH study in which patients were randomized to different lengths of neoadjuvant induction anti-HER2 therapy with lapatinib and trastuzumab followed by weekly paclitaxel plus the anti-HER2 therapy, and in estrogen receptor (ER)-positive patients, with or without concurrent endocrine therapy. The use of endocrine therapy did not affect the response; comprehensive pathological complete response (CpCR) plus ypN0 rate was 57.6% and 30.3% in ER-negative and ER-positive patients, respectively. After surgery, patients received an anthracycline-based regimen based on physician’s choice, followed by trastuzumab for 1 year, and in ER-positive patients, endocrine therapy for 5 years….”

3) Results in table 1 suggest that the duration of induction therapy and the addition of endocrine did not impact pCR rates in the two strata but you should say that explicitly. Also, you should state explicitly that these randomizations did not impact DFS or other long-term outcomes (assuming this is the case). 

Response

We have revised the text as follows.

3.1. Patients

“Table 1 shows the patients’ characteristics. The median age was 53 years (range: 26–70 years) and the duration of follow-up was 2074 days (range: 63–2425 days). CpCRypN0 rate was 65.9% and 58.3% in groups A and B (ER-negative cohort), and 31.7%, 33.3%, and 37.5% in groups C-E (ER-positive cohort), showing no difference within each cohort associated with the duration of neoadjuvant induction therapy and/or the addition of endocrine therapy.”

3.2. Survival Outcomes

“By subgroups at randomization, the 5-year DFS was 88.3% and 89.4% in groups A and B (ER-negative cohort), and 85.4%, 86.9%, and 89.3% in groups C-E (ER-positive cohort). The 5-year DDFS was 95.3% and 93.7% in groups A and B (ER-negative cohort), and 90.2%, 94.7% and 94.6% in groups C-E (ER-positive cohort). The 5-year OS was 97.6 and 93.5%, and 97.6%, 92.1%, and 97.3% in groups C-E (ER-positive cohort). Therefore, no substantial difference in the survival outcomes was noted in both cohorts associated with the duration of neoadjuvant induction therapy and/or the addition of endocrine therapy.”

4) In Table 1 you should break down receipt of post-op radiation separately between the BCS vs. mastectomy patients; as presented it looks like only half on the BCS patients received radiation. Also, in that same table, what does QpCRypN0 refer to? Actually, aside from the pCR rates, the rest of the table could be moved to the Supplement, as the data is really not necessary for your current analysis. 

Response

Actually, Table 1 shows that, of the patients who had breast conservation (N=130), 125 patients received postoperative radiation (7 patients with regional lymph node irradiation plus 118 patients without regional lymph node irradiation).

QpCRypN0 refers to CpCR(ypT0/is) plus near pCR. As you pointed out, it is not relevant for the current analysis, so we have omitted data from Table 1.

5) The absence of a significant difference in DFS between pCR and non-pCR among the ER- cohort is, presumably, a result of the small size of the cohort - would cite statistics and eliminate Figure 4. Similarly, once you've shown results for DFS, you can simply cite statistics for DDFS and OS - Figures 5 and 6 are unnecessary (again you could put these in the Supplement but they don't improve the manuscript).

Response

As suggested, we have moved Figure 4 to Supplement and omitted Figures 5 and 6.

6) Table 2 is confusing - would revise or eliminate and simply state the percentages of patients who did or did not receive adjuvant chemotherapy by cohort and pCR, especially since it didn't make a difference . Would keep Figure 7A but eliminate 7B and 7C. 

Response

We have revised Table 2 as below.

Table 2. Proportion of patients who received adjuvant anthracycline chemotherapy

Achieved CpCRypN0

Residual invasive disease

Total

ER-negative

29.8% (17/57)

45.7% (16/35)

35.9% (33/92)

ER-positive

34.1% (14/41)

71.4% (55/77)

58.5% (69/118)

Total

31.6% (31/98)

63.4% (71/112)

48.6% (102/210 a)

ER, estrogen receptor. a From the total 212 patients, one patient who withdrew consent during the follow-up period (group E) and 1 patient who did not undergo surgery (group E) were excluded.

As suggested, we have omitted Figures 7B and 7C.

7) Would mention outcomes in the lower risk subgroup but eliminate Figure 8. 

Response

We have moved Figure 8A as Supplement, and omitted Figures 8B and 8C.

8) in your discussion of outcomes in the ER+ cohort and the role of 'crosstalk' in determining response to neoadjuvant therapy +/- endocrine therapy, you should mention the negative results from NSABP B-52.

Response

As suggested, we have revised the relevant part in the Discussion as follows.

“By contrast, HER2-positive/hormone receptor-positive tumors are generally less dependent on HER2 signaling than HER2-positive/hormone receptor-negative tumors. HER2-positive/hormone receptor-positive tumors cause upregulation of hormone receptor signaling as an adaptive mechanism of cell survival, which may lead to partial resistance to anti-HER2 therapy [23]. In this subtype, risk of recurrence remains for a longer time. The crosstalk between ER and HER2 receptor signaling may be an important contributor to the development of resistance to therapies against the ER pathway [24]. Various growth factor receptor and other intracellular signaling pathways may be activated or overexpressed in breast cancer, especially in endocrine-resistant cells. It is suggested that concurrent targeting of ER and HER2 may improve treatment efficacy and overcome ER-mediated resistance. Clinically, the NRG Oncology/NSABP B-52 study showed that the addition of endocrine therapy to neoadjuvant therapy (docetaxel, carboplatin, trastuzumab, and pertuzumab) improved pCR rates numerically, but the improvement was not statistically significant [25]. Likewise, in the present study, CpCR rate after neoadjuvant treatment and long-term survival outcomes were similar among ER-positive patients, regardless of the addition of endocrine therapy to neoadjuvant anti-HER2 therapy plus paclitaxel at randomization. Moreover, generally, a pCR can be an important prognostic factor for luminal type B breast cancers, whereas the effect of neoadjuvant chemotherapy is not a good indicator for luminal type A [26-28]. A mixture of these two types may explain why there is little difference in prognosis be-tween patients with and without a pCR in this cohort.

Additional reference

  1. Rimawi, M.F.; Cecchini, R.S.; Rastogi, P.; Geyer, C.E., Jr; Fehrenbacher, L.; Stella, P.J.; Dayao, Z.; Rabinovitch, R.; Dyar, S.H.; Flynn, P.J.; Baez-Diaz, L.; Paik, S.; Swain, S.M.; Mamounas, E.P.; Osborne, C.K.; Wolmark, N. A phase III trial evaluating pCR in patients with HR+, HER2-positive breast cancer treated with neoadjuvant docetaxel, carboplatin, trastuzumab, and pertuzumab (TCHP) +/- estrogen deprivation: NRG Oncology/NSABP B-52. In Proceedings of the 2016 San Antonio Breast Cancer Symposium, San Antonio, TX, USA, 6-10, Dec, 2016; AACR; Cancer. Res. 2017,77(4 Suppl), Abstract S3-06.

9) You correctly point out that the ability to draw conclusions from the lack of benefit from adjuvant chemotherapy is limited since this was not randomized, but your data do not support your conclusion that it can be omitted only in patients who achieve pCR.  

Response

We have revised the Discussion as follows.

“Because of concerns for anthracycline-induced cardiac toxicity, many physicians are willing to avoid anthracycline-containing regimens in patients who achieve a pCR or, even in patients who do not have a pCR, when the residual tumor is small (especially pN0 cases). Our findings suggested a possible omission of anthracycline in patients who achieve CpCRypN0 after neoadjuvant treatment. However, a prospective, randomized study is necessary to validate these results. Moreover, in patients who had residual invasive tumor after neoadjuvant treatment, the survival outcomes were less favorable in both cases with and without adjuvant anthracycline therapy. In such cases, more intensive adjuvant chemotherapy regimens may be needed; novel antibody-drug conjugate agents including trastuzumab emtansine (T-DM1) [31] and trastuzumab deruxtecan (DS-8201a) [32,33] are being investigated and promising results have been obtained.”

Additional references

  1. von Minckwitz, G.; Huang, C.S.; Mano, M.S.; Loibl, S.; Mamounas, E.P.; Untch, M.; Wolmark N.; Rastogi, P.; Schneeweiss, A.; Redondo, A.; Fischer, H.H.; Jacot, W. et al. for the KATHERINE Investigators. Trastuzumab emtansine for residual invasive HER2-positive breast cancer. N. Engl. J. Med. 2019, 380, 617-628,
  2. Tamura, K.; Tsurutani, J.; Takahashi, S.; Iwata, H.; Krop, I.E.; Redfern, C.; Sagara, Y.; Doi, T.; Park, H.; Murthy, R.K.; Redman R.A.; Jikoh, T.; Lee, C.; Sugihara, M.; Shahidi, J.; Yver, A.; Modi, S. Trastuzumab deruxtecan (DS-8201a) in patients with advanced HER2-positive breast cancer previously treated with trastuzumab emtansine: a dose-expansion, phase 1 study. Lancet. Oncol. 2019, 20, 816-826.
  3. Modi, S.; Saura, C.; Yamashita, T.; Park, Y.H.; Kim, S.B.; Tamura, K.; Andre, F.; Iwata, H.; Ito, Y.; Tsurutani, J.; Sohn, J.; Denduluri, N. et al. for the DESTINY-Breast01 Investigators.Trastuzumab deruxtecan in previously treated HER2-positive breast cancer. N. Engl. J. Med. 2020, 382, 610-621.

Reviewer 2 Report

The authors present survival analyses from a previously published Neoadjuvant trial in HER2+ BC where double blockade with Trastuzumab + Lapatinib + chemotherapy with paclitaxel, with our without ET was administered. Results are in line with most of the published literature in the field and add significant exploratory data regarding the role of Anthracyclines in this setting. I have no major concerns regarding the publication of this manuscript, however I hereby report some minor comments or issues that I would like the authors to address/solve. 

1) Page 3, Line 123 please avoid  repetitions of “treatment”. “Therapy” could be used alternatively or directly “response to Neoadjuvant treatment,”.

2) The Simple Summary is not in layman’s terms. The authors should try to simplify more the key messages of their study in this section. 

3) With respect to patients receiving ET in the Neoadjuvant setting, it is not clear whether or not they received a total of 5 years ET (including both Neo and adj part) or 5 years adjuvant ET + (for groups D and E) several months of Neoadjuvant ET. This should be better clarified. Moreover, did the authors notice any difference in survival outcomes for the HR+ cohort based on whether or not ET was administered also in Neoadjuvant setting? If not explored, the authors might consider providing this exploratory analysis in the section 3.4. 

Author Response

Manuscript ID: cancers-1289943

Long-term Outcomes of a Randomized Study of Neoadjuvant Induction Dual HER2 Blockade with Trastuzumab and Lapa-tinib Followed by Weekly Paclitaxel plus Dual HER2 Blockade for HER2-Positive Primary Breast Cancer (Neo-Lath Study)

Dear Reviewer,

Thank you for considering the above manuscript for publication in Cancers. Your comments were highly insightful and enabled us to improve the quality of the manuscript. In the following pages, please find our point-by point response to each of the comments.

In the manuscript file, revisions are shown using the Microsoft Word tracked changes feature. We trust that the revisions in the manuscript will be sufficient to make our manuscript suitable for publication in Cancers.

Comments from Reviewer 2:

The authors present survival analyses from a previously published Neoadjuvant trial in HER2+ BC where double blockade with Trastuzumab + Lapatinib + chemotherapy with paclitaxel, with our without ET was administered. Results are in line with most of the published literature in the field and add significant exploratory data regarding the role of Anthracyclines in this setting. I have no major concerns regarding the publication of this manuscript, however I hereby report some minor comments or issues that I would like the authors to address/solve. 

1) Page 3, Line 123 please avoid  repetitions of “treatment”. “Therapy” could be used alternatively or directly “response to Neoadjuvant treatment,”.

Response

We have revised the relevant text as suggested.

“After surgery, patients received an anthracycline-based regimen depending on the physician’s choice and response to neoadjuvant treatment (this regimen could be omitted in cases of pCR and N0).”

2) The Simple Summary is not in layman’s terms. The authors should try to simplify more the key messages of their study in this section. 

We have revised the Simple Summary as follows.

Simple Summary:

“We conducted the Neo-LaTH study, in which patients with HER2-positive breast cancer were randomized to different lengths of neoadjuvant in-duction anti-HER2 therapy with lapatinib and trastuzumab followed by weekly paclitaxel plus anti-HER2 therapy, and in estrogen receptor-positive patients, with or without concurrent endocrine therapy. Here, we report the survival outcomes. The duration of neoadjuvant induction therapy and/or the addition of endocrine therapy at randomization did not affect the pathological complete response (CpCR) rate after neoadjuvant treatment and long-term outcomes. The 5-year disease-free survival rate was significantly higher in patients who had a CpCR plus ypN0 after neoadjuvant treatment than in those who did not (91.7% vs 85.1%; P=0.0387). The stratified analysis showed better survival outcomes in CpCRypN0 patients than non-CpCRypN0 patients, regardless of use of adjuvant anthracycline therapy. Favorable survival outcomes, regardless of adjuvant anthracycline, were particularly noted in patients with small size and clinically node-negative tumor.”

3) With respect to patients receiving ET in the Neoadjuvant setting, it is not clear whether or not they received a total of 5 years ET (including both Neo and adj part) or 5 years adjuvant ET + (for groups D and E) several months of Neoadjuvant ET. This should be better clarified. Moreover, did the authors notice any difference in survival outcomes for the HR+ cohort based on whether or not ET was administered also in Neoadjuvant setting? If not explored, the authors might consider providing this exploratory analysis in the section 3.4. 

Response

Patients received 5-year adjuvant endocrine therapy, regardless of the length of the neoadjuvant endocrine therapy. This is now stated in the Methods (2.1. Trial design).

Regarding the survival outcomes, the addition of endocrine therapy did not affect the response in ER-positive patients. We have added the relevant information as follows.

3.1. Patients

“Table 1 shows the patients’ characteristics. The median age was 53 years (range: 26–70 years) and the duration of follow-up was 2074 days (range: 63–2425 days). CpCRypN0 rate was 65.9% and 58.3% in groups A and B (ER-negative cohort), and 31.7%, 33.3%, and 37.5% in groups C-E (ER-positive cohort), showing no difference within each cohort associated with the duration of neoadjuvant induction therapy and/or the addition of endocrine therapy.”

3.2. Survival Outcomes

“By subgroups at randomization, the 5-year DFS was 88.3% and 89.4% in groups A and B (ER-negative cohort), and 85.4%, 86.9% and 89.3% in groups C-E (ER-positive cohort). The 5-year DDFS was 95.3% and 93.7% in groups A and B (ER-negative cohort), and 90.2%, 94.7%, and 94.6% in groups C-E (ER-positive cohort). The 5-year OS was 97.6 and 93.5%, and 97.6%, 92.1%, and 97.3% in groups C-E (ER-positive cohort). Therefore, no substantial difference in the survival outcomes was noted in both cohorts associated with the duration of neoadjuvant induction therapy and/or the addition of endocrine therapy.”

Round 2

Reviewer 1 Report

Thank you for your revisions in response to previous comments and suggestions by reviewers - the manuscript is much clearer now. Can you comment on why patients with ER-/HER2+ cancers who did not achieve pCR were less likely (16/35)  than those with ER+/HER2+ cancers (55/77) to receive adjuvant anthracycline? - I would have expected the opposite result unless many of the ER-/HER2+ patients who did not achieve pCR had minimal residual disease, so their doctors chose not to give them more chemo.